# Module-wise Training of Neural Networks via the Minimizing Movement Scheme

**Skander Karkar**
Criteo, Sorbonne Université

**Ibrahim Ayed**
Sorbonne Université, Thales

**Emmanuel de Bézenac**
ETH Zurich

**Patrick Gallinari**
Criteo, Sorbonne Université

## Abstract

Greedy layer-wise or module-wise training of neural networks is compelling in constrained and on-device settings where memory is limited, as it circumvents a number of problems of end-to-end back-propagation. However, it suffers from a stagnation problem, whereby early layers overfit and deeper layers stop increasing the test accuracy after a certain depth. We propose to solve this issue by introducing a module-wise regularization inspired by the minimizing movement scheme for gradient flows in distribution space. We call the method TRGL for Transport Regularized Greedy Learning and study it theoretically, proving that it leads to greedy modules that are regular and that progressively solve the task. Experimentally, we show improved accuracy of module-wise training of various architectures such as ResNets, Transformers and VGG, when our regularization is added, superior to that of other module-wise training methods and often to end-to-end training, with as much as $60\%$ less memory usage.

## 1 Introduction

End-to-end backpropagation is the standard training method of neural networks. However, it requires storing the whole model and computational graph during training, which requires large memory consumption. It also prohibits training the layers in parallel. Dividing the network into modules, a module being made up of one or more layers, accompanied by auxiliary classifiers, and greedily solving module-wise optimization problems sequentially (i.e. one after the other fully) or in parallel (i.e. at the same time batch-wise), consumes much less memory than end-to-end training as it does not need to store as many activations, and when done sequentially, only requires loading and training one module (so possibly one layer) at a time. Module-wise training has therefore been used in constrained settings in which end-to-end training can be impossible such as training on mobile devices [58, 57] and dealing with very large whole slide images [65]. When combined with batch buffers, parallel module-wise training also allows for parallel training of the modules [8]. Despite its simplicity, module-wise training has been recently shown to scale well [8, 47, 60, 45], outperforming more complicated alternatives to end-to-end training such as synthetic [33, 14] and delayed [32, 31] gradients, while having superior memory savings.

In a classification task, module-wise training splits the network into successive modules, a module being made up of one or more layers. Each module takes as input the output of the previous module, and each module has an auxiliary classifier so that a local loss can be computed, with backpropagation happening only inside the modules and not between them (see Figure 1 below).

The main drawback of module-wise training is the well-documented *stagnation problem* observed in [43, 7, 60, 47], whereby early modules overfit and learn more discriminative features than end-to-end

37th Conference on Neural Information Processing Systems (NeurIPS 2023).

training, destroying task-relevant information, and deeper modules don't improve the test accuracy significantly, or even degrade it, which limits the deployment of module-wise training. We further highlight this phenomenon in Figures 2 and 3 in Section 4.4. To tackle this issue, InfoPro [60] propose to maximize the mutual information that each module keeps with the input, in addition to minimizing the loss. [7] make the auxiliary classifier deeper and Sedona [47] make the first module deeper. These last two methods lack a theoretical grounding, while InfoPro requires a second auxiliary network for each module besides the classifier. We propose a different perspective, leveraging the analogy between residual connections and the Euler scheme for ODEs [61]. To preserve input information, we minimize the kinetic energy of the modules along with the training loss. Intuitively, this forces the modules to change their input as little as possible. We leverage connections with the theories of gradient flows in distribution space and optimal transport to analyze our method theoretically.

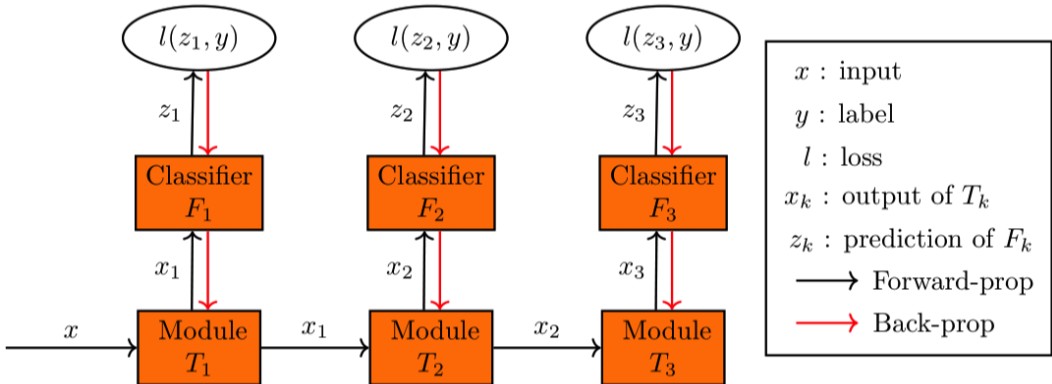

Figure 1: Module-wise training.

Our approach is particularly well-adapted to networks that use residual connections such as ResNets [27, 28], their variants (e.g. ResNeXt [62], Wide ResNet [63], EfficientNet [56] and MobileNetV2 [48]) and vision transformers that are made up essentially of residual connections [39, 17], but is immediately usable on any network where many layers have the same input and output dimension such as VGG [52]. Our contributions are the following:

- We propose a new method for module-wise training. Being a regularization, it is lighter than many recent state-of-the-art methods (PredSim [45], InfoPro [60]) that train another auxiliary network besides the auxiliary classifier for each module.

- We theoretically justify our method, proving that it is a transport regularization that forces the module to be an optimal transport map making it more regular and stable. We also show that it amounts to a discretization of the gradient flow of the loss in probability space, which means that the modules progressively minimize the loss and explains why the method avoids the accuracy collapse observed in module-wise training.

- Experimentally, we consistently improve the test accuracy of module-wise trained networks (ResNets, VGG and Swin-Transformer) beating 8 other methods, in sequential and parallel module-wise training, and also in *multi-lap sequential* training, a variant of sequential module-wise training that we introduce and that performs better in many cases. In particular, our regularization makes parallel module-wise training superior or comparable in accuracy to end-to-end training, while consuming $10\%$ to $60\%$ less memory.

## 2   Transport-regularized module-wise training

The typical setting of (sequential) module-wise training for minimizing a loss $L$, is, given a dataset $\mathcal{D}$, to solve one after the other, for $1 \leq k \leq K$, Problems

$$(T_k, F_k) \in \arg\min_{T,F} \sum_{x \in \mathcal{D}} L(F, T \circ G_{k-1}(x)) \tag{1}$$

where $G_k = T_k \circ ... \circ T_1$ for $1 \leq k \leq K$, $G_0 = \texttt{id}$, $T_k$ is the module (one or many layers) and $F_k$ is an auxiliary classifier. Module $T_k$ receives the output of module $T_{k-1}$, and auxiliary classifier $F_k$ computes the prediction from the output of $T_k$ so the loss can be computed. The inputs are $x$ and $L$ has access to their labels $y$ to calculate the loss. i.e. $L(F, T \circ G_{k-1}(x)) = l(F \circ T \circ G_{k-1}(x), y)$ where $l$ is a machine learning loss such as cross-entropy. See Figure 1. The final network trained this way is $F_K \circ G_K$. But, at inference, we can stop at any depth $k$ and use $F_k \circ G_k$ if it performs better. Indeed, an intermediate module often performs as well or better than the last module because of the early overfitting and subsequent stagnation or collapse problem of module-wise training [43, 7, 60, 47].

We propose below in (2) a regularization that avoids the destruction of task-relevant information by the early modules by forcing them to minimally modify their input. Proposition 2.2 proves that by using our regularization (2), we are indeed making the modules build upon each other to solve the task, which is the property we desire in module-wise training, as the modules now act as successive proximal optimization steps in the *minimizing movement scheme* optimization algorithm for maximizing the separability of the data representation. The background on optimal transport (OT), gradient flows and the minimizing movement scheme is in Appendices A and B.

## 2.1 Method statement

To keep greedily-trained modules from overfitting and destroying information needed later, we penalize their kinetic energy to force them to preserve the geometry of the problem as much as possible. If each module is a single residual block (that is a function $T = \texttt{id} + r$, which includes many transformer architectures [39, 17]), its kinetic energy is simply the squared norm of its residue $r = T - \texttt{id}$, which we add to the loss $L$ in the target of the greedy problems (1). All layers that have the same input and output dimension can be rewritten as residual blocks and the analysis applies to a large variety of architectures such as VGG [52]. Given $\tau > 0$, we now solve, for $1 \leq k \leq K$, Problems

$$(T_k^\tau, F_k^\tau) \in \arg\min_{T,F} \sum_{x \in \mathcal{D}} L(F, T \circ G_{k-1}^\tau(x)) + \frac{1}{2\tau} \|T \circ G_{k-1}^\tau(x) - G_{k-1}^\tau(x)\|^2 \qquad (2)$$

where $G_k^\tau = T_k^\tau \circ .. \circ T_1^\tau$ for $1 \leq k \leq K$ and $G_0^\tau = \texttt{id}$. The final network is $F_K^\tau \circ G_K^\tau$. Intuitively, this biases the modules towards moving the points as little as possible, thus at least keeping the performance of the previous module. Residual connections are already biased towards small displacements and this bias is desirable and should be encouraged [35, 64, 26, 15, 36]. But the method can be applied to any module where $T(x)$ and $x$ have the same dimension so that $T(x) - x$ can be computed.

To facilitate the theoretical analysis, we rewrite the method in a more general formulation using data distribution $\rho$, which can be discrete or continuous, and the distribution-wide loss $\mathcal{L}$ that arises from the point-wise loss $L$. Then Problem (2) is equivalent to Problem

$$(T_k^\tau, F_k^\tau) \in \arg\min_{T,F} \mathcal{L}(F, T_\sharp \rho_k^\tau) + \frac{1}{2\tau} \int_\Omega \|T(x) - x\|^2 \, \mathrm{d}\rho_k^\tau(x) \qquad (3)$$

with $\rho_{k+1}^\tau = (T_k^\tau)_\sharp \rho_k^\tau$, $\rho_1^\tau = \rho$ and $\mathcal{L}(F, T_\sharp \rho_k^\tau) = \int L(F, T(x)) \, \mathrm{d}\rho_k^\tau(x) = \int L(F, z) \, \mathrm{d}T_\sharp \rho_k^\tau(x)$.

## 2.2 Link with the minimizing movement scheme

We now formulate our main result: solving Problems (3) is equivalent to following a *minimizing movement scheme (MMS)* [50] in distribution space for minimizing $\mathcal{Z}(\mu) := \min_F \mathcal{L}(F, \mu)$, which is the loss of the best classifier. If we are limited to linear classifiers, $\mathcal{Z}(\rho_k^\tau)$ is the linear separability of the representation $\rho_k^\tau$ at module $k$ of the data distribution $\rho$. The MMS, introduced in [24, 23], is a metric counterpart to Euclidean gradient descent for minimizing functionals over distributions. In our case, $\mathcal{Z}$ is the functional we want to minimize. We define the MMS below in Definition 2.1

The distribution space we work in is the metric Wasserstein space $\mathbb{W}_2(\Omega) = (\mathcal{P}(\Omega), W_2)$, where $\Omega \subset \mathbb{R}^d$ is a convex compact set, $\mathcal{P}(\Omega)$ is the set of probability distributions over $\Omega$ and $W_2$ is the Wasserstein distance over $\mathcal{P}(\Omega)$ derived from the optimal transport problem with Euclidean cost:

$$W_2^2(\alpha, \beta) = \min_{T \text{ s.t. } T_\sharp \alpha = \beta} \int_\Omega \|T(x) - x\|^2 \, \mathrm{d}\alpha(x) \qquad (4)$$

where we assume that $\partial\Omega$ is negligible and that the distributions are absolutely continous.

**Definition 2.1.** Given $\mathcal{Z} : \mathbb{W}_2(\Omega) \to \mathbb{R}$, and starting from $\rho_1^\tau \in \mathcal{P}(\Omega)$, the Minimizing Movement Scheme (MMS) takes proximal steps for minimizing $\mathcal{Z}$. It is s given by

$$\rho_{k+1}^\tau \in \arg \min_{\rho \in \mathcal{P}(\Omega)} \mathcal{Z}(\rho) + \frac{1}{2\tau} W_2^2(\rho, \rho_k^\tau) \tag{5}$$

The MMS (5) can be seen as a non-Euclidean implicit Euler step for following the gradient flow of $\mathcal{Z}$, and $\rho_k^\tau$ converges to a minimizer of $\mathcal{Z}$ under some conditions (see the end of this section).

So under the mentioned assumptions on $\Omega$ and absolute continuity of the distributions, we have that Problems (3) are equivalent to the minimizing movement scheme (5):

**Proposition 2.2.** *The distributions $\rho_{k+1}^\tau = (T_k^\tau)_\sharp \rho_k^\tau$, where the functions $T_k^\tau$ are found by solving (3) and $\rho_1^\tau = \rho$ is the data distribution, coincide with the MMS (5) for $\mathcal{Z} = \min_F \mathcal{L}(F, .)$.*

*Proof.* The minimizing movement scheme (5) is equivalent to taking $\rho_{k+1}^\tau = (T_k^\tau)_\sharp \rho_k^\tau$ where

$$T_k^\tau \in \arg \min_{T:\Omega \to \Omega} \mathcal{Z}(T_\sharp \rho_k^\tau) + \frac{1}{2\tau} W_2^2(T_\sharp \rho_k^\tau, \rho_k^\tau) \tag{6}$$

under conditions that guarantee the existence of a transport map between $\rho_k^\tau$ and any other measure, and absolute continuity of $\rho_k^\tau$ suffices, and the loss can ensure that $\rho_{k+1}^\tau$ is also absolutely continuous. Among the functions $T_k^\tau$ that solve problem (6), is the optimal transport map from $\rho_k^\tau$ to $\rho_{k+1}^\tau$. To solve specifically for this optimal transport map, we have to solve the equivalent Problem

$$T_k^\tau \in \arg \min_T \mathcal{Z}(T_\sharp \rho_k^\tau) + \frac{1}{2\tau} \int_\Omega \|T(x) - x\|^2 \, d\rho_k^\tau(x) \tag{7}$$

Problems (6) and (7) have the same minimum value, but the minimizer of (7) is now an optimal transport map between $\rho_k^\tau$ and $\rho_{k+1}^\tau$. This is immediate from the definition (4) of the $W_2$ distance. Equivalently minimizing first over $F$ and then over $T$ in (3), it follows from the definition of $\mathcal{Z}$ that Problems (3) and (7) are equivalent, which concludes. $\square$

Since we solve Problems (3) over neural networks, their representation power shown by universal approximation theorems [13, 29] is important to get close to equivalence between (5) and (3), as we need to approximate an optimal transport map. We also know that the training of each module, if it is shallow, converges [5, 6, 34, 22, 18].

If $\mathcal{Z}$ is lower-semi continuous then Problems (5) always admit a solution because $\mathcal{P}(\Omega)$ is compact. If $\mathcal{Z}$ is also $\lambda$-geodesically convex for $\lambda > 0$, we have convergence of $\rho_k^\tau$ as $k \to \infty$ and $\tau \to 0$ to a minimizer of $\mathcal{Z}$, potentially under more technical conditions (see Appendix B). Even though a machine learning loss will usually not satisfy these conditions, this analysis offers hints as to why our method avoids in practice the problem of stagnation or collapse in performance of module-wise training as $k$ increases, as we are making proximal local steps in Wasserstein space to minimize the loss. This convergence discussion also suggests taking $\tau$ as small as possible and many modules.

## 2.3 Regularity result

As a secondary result, we show that Problem (3) has a solution and that the solution module $T_k^\tau$ is an optimal transport map between its input and output distributions, which means that it comes with some regularity. [36] show that these networks generalize better and overfit less in practice. We assume that the minimization in $F$ is over a compact set $\mathcal{F}$, that $\rho_k^\tau$ is absolutely continuous, that $\mathcal{L}$ is continuous and non-negative, that $\Omega$ is convex and compact and that $\partial\Omega$ is negligible.

**Proposition 2.3.** *Problem (3) has a minimizer $(T_k^\tau, F_k^\tau)$ such that $T_k^\tau$ is an optimal transport map. And for any minimizer $(T_k^\tau, F_k^\tau)$, $T_k^\tau$ is an optimal transport map.*

The proof is in Appendix C. OT maps have regularity properties under some boundedness assumptions. Given Theorem A.1 in Appendix A taken from [20], $T_k^\tau$ is $\eta$-Hölder continuous almost everywhere and if the optimization algorithm we use to solve the discretized problem (2) returns an approximate solution pair $(\tilde{F}_k^\tau, \tilde{T}_k^\tau)$ such that $\tilde{T}_k^\tau$ is an $\epsilon$-optimal transport map, i.e. $\|\tilde{T}_k^\tau - T_k^\tau\|_\infty \leq \epsilon$, then we have (using the triangle inequality) the following stability property of the module $\tilde{T}_k^\tau$:

$$\|\tilde{T}_k^\tau(x) - \tilde{T}_k^\tau(y)\| \leq 2\epsilon + C\|x - y\|^\eta \tag{8}$$

for almost every $x, y \in \text{supp}(\rho_k^\tau)$ and $C > 0$. Composing these stability bounds on $T_k^\tau$ and $\tilde{T}_k^\tau$ allows to get bounds for the composition networks $G_k^\tau$ and $\tilde{G}_k^\tau = \tilde{T}_k^\tau \circ .. \circ \tilde{T}_1^\tau$.

To summarize Section 2, the transport regularization makes each module more regular and it allows the modules to build on each other as $k$ increases to solve the task, which is the property we desire.

# 3 Practical implementation

## 3.1 Multi-block modules

For simplicity, we presented in (2) the case where each module is a single residual block. However, in practice, we often split the network into modules that are made-up of many residual blocks each. We show here that regularizing the kinetic energy of such modules still amounts to a transport regularization, which means that the theoretical results in Propositions 2.2 and 2.3 still apply.

If each module $T_k$ is made up of $M$ residual blocks, i.e. applies $x_{m+1} = x_m + r_m(x_m)$ for $0 \le m < M$, then its total discrete kinetic energy for a single data point $x_0$ is the sum of its squared residue norms $\sum \|r_m(x_m)\|^2$, since a residual network can be seen as a discrete Euler scheme for an ordinary differential equation [61] with velocity field $r$:

$$x_{m+1} = x_m + r_m(x_m) \longleftrightarrow \partial_t x_t = r_t(x_t) \tag{9}$$

and $\sum \|r_m(x_m)\|^2$ is then the discretization of the total kinetic energy $\int_0^1 \|r_t(x)\|^2 \, \mathrm{d}t$ of the ODE. If $\psi_m^x$ denotes the position of a point $x$ after $m$ residual blocks, then regularizing the kinetic energy of multi-block modules now means solving

$$(T_k^\tau, F_k^\tau) \in \arg\min_{T,F} \sum_{x \in \mathcal{D}} (L(F, T(G_{k-1}^\tau(x)) + \frac{1}{2\tau} \sum_{m=0}^{M-1} \|r_m(\psi_m^x)\|^2) \tag{10}$$

$$\text{s.t. } T = (\mathrm{id} + r_{M-1}) \circ ... \circ (\mathrm{id} + r_0), \ \psi_0^x = G_{k-1}^\tau(x), \psi_{m+1}^x = \psi_m^x + r_m(\psi_m^x)$$

where $G_k^\tau = T_k^\tau \circ .. \circ T_1^\tau$ for $1 \le k \le K$ and $G_0^\tau = \mathrm{id}$. We also minimize this sum of squared residue norms instead of $\|T(x) - x\|^2$ (the two no longer coincide) as it works better in practice, which we assume is because it offers a more localized control of the transport. As expressed in (9), a residual network can be seen as an Euler scheme of an ODE and Problem (10) is then the discretization of

$$(T_k^\tau, F_k^\tau) \in \arg\min_{T,F} \ \mathcal{L}(F, T_\sharp \rho_k^\tau) + \frac{1}{2\tau} \int_0^1 \|v_t\|_{L^2((\phi_t)_\sharp \rho_k^\tau)}^2 \, \mathrm{d}t \tag{11}$$

$$\text{s.t. } T = \phi_1^\cdot, \ \partial_t \phi_t^x = v_t(\phi_t^x), \ \phi_0^\cdot = \mathrm{id}$$

where $\rho_{k+1}^\tau = (T_k^\tau)_\sharp \rho_k^\tau$ and $r_m$ is the discretization of vector field $v_t$ at time $t = m/M$. Here, distributions $\rho_k^\tau$ are pushed forward through the maps $T_k^\tau$ which correspond to the flow $\phi$ at time $t = 1$ of the kinetically-regularized velocity field $v_t$. We recognize in the second term in the target of (11) the optimal transport problem in its dynamic formulation (15) from [9], and given the equivalence between the Monge OT problem (4) and the dynamic OT problem (15) in Appendix A, Problem (11) is in fact equivalent to the original continuous formulation (3), and the theoretical results in Section 2 follow immediately (see also the proof of Proposition 2.3 in Appendix C).

## 3.2 Solving the module-wise problems

The module-wise problems can be solved in two ways. One can completely train each module with its auxiliary classifier for $N$ epochs before training the next module, which receives as input the output of the previous trained module. We call this *sequential* module-wise training. But we can also do this batch-wise, i.e. do a complete forward pass on each batch but without a full backward pass, rather a backward pass that only updates the current module $T_k^\tau$ and its auxiliary classifier $F_k^\tau$, meaning that $T_k^\tau$ forwards its output to $T_{k+1}^\tau$ immediately after it computes it. We call this *parallel* module-wise training. It is called *decoupled* greedy training in [8], which shows that combining it with batch buffers solves all three locking problems and allows a linear training parallelization in the depth of the network. We propose a variant of sequential module-wise training that we call *multi-lap sequential* module-wise training, in which instead of training each module for $N$ epochs, we train each module from the first to the last sequentially for $N/R$ epochs, then go back and train from the first module to

the last for $N/R$ epochs again, and we do this for $R$ laps. For the same total number of epochs and training time, and the same advantages (loading and training one module at a time) this provides a non-negligible improvement in accuracy over normal sequential module-wise training in most cases, as shown in Section 4. Despite our theoretical framework being that of sequential module-wise training, our method improves the test accuracy of all three module-wise training regimes.

### 3.3 Varying the regularization weight

The discussion in Section 2.2 suggests taking a fixed weight $\tau$ for the transport cost that is as small as possible. However, instead of using a fixed $\tau$, we might want to vary it along the depth $k$ to further constrain with a smaller $\tau_k$ the earlier modules to avoid that they overfit or the later modules to maintain the accuracy of earlier modules. We might also want to regularize the network further in earlier epochs when the data is more entangled. We propose in Appendix D to formalize this varying weight $\tau_{k,i}$ across modules $k$ and SGD iterations $i$ by using a scheme inspired by the method of multipliers to solve Problems (2) and (10). However, it works best in only one experiment in practice. The observed dynamics of $\tau_{k,i}$ suggest simply finding a fixed value of $\tau$ that is multiplied by 2 for the second half of the network, which works best in all the other experiments (see Appendix E).

## 4 Experiments

We call our method TRGL for Transport-Regularized Greedy Learning. For the auxiliary classifiers, we use the architecture from DGL [7, 8], that is a convolution followed by an average pooling and a fully connected layer, which is very similar to that used by InfoPro [60], except for the Swin Transformer where we use a linear layer. We call vanilla greedy module-wise training with the same architecture but without our regularization VanGL, and we include its results in all tables for ablation study purposes. The code is available at `github.com/block-wise/module-wise` and implementation details are in Appendix E.

### 4.1 Parallel module-wise training

To compare with other methods, we focus first on parallel training, as it performs better than sequential training and has been more explored recently. The first experiment is training in parallel 3 residual architectures and a VGG-19 [52] divided into 4 modules of equal depth on TinyImageNet. We compare in Table 1 our results in this setup to three of the best recent parallel module-wise training methods: DGL [8], PredSim [45] and Sedona [47], from Table 2 in [47]. We find that our TRGL has a much better test accuracy than the three other methods, especially on the smaller architectures. It also performs better than end-to-end training on the three ResNets. Parallel TRGL in this case with 4 modules consumes 10 to 21% less memory than end-to-end training (with a batch size of 256).

Table 1: Test accuracy of parallel TRGL with 4 modules (average and 95% confidence interval over 5 runs) on TinyImageNet, compared to DGL, PredSim, Sedona and E2E from Table 2 in [47], with memory saved compared to E2E as a percentage of E2E memory consumption in red.

| Architecture | Parallel VanGL | Parallel TRGL (ours) | PredSim | DGL | Sedona | E2E |
|---|---|---|---|---|---|---|
| VGG-19 | $56.17 \pm 0.29$ ($\downarrow 27\%$) | $\mathbf{57.28} \pm 0.20$ ($\downarrow 21\%$) | 44.70 | 51.40 | 56.56 | 58.74 |
| ResNet-50 | $58.43 \pm 0.45$ ($\downarrow 26\%$) | $\mathbf{60.30} \pm 0.58$ ($\downarrow 20\%$) | 47.48 | 53.96 | 54.40 | 58.10 |
| ResNet-101 | $63.64 \pm 0.30$ ($\downarrow 24\%$) | $\mathbf{63.71} \pm 0.40$ ($\downarrow 11\%$) | 53.92 | 53.80 | 59.12 | 62.01 |
| ResNet-152 | $63.87 \pm 0.16$ ($\downarrow 21\%$) | $\mathbf{64.23} \pm 0.14$ ($\downarrow 10\%$) | 51.76 | 57.64 | 64.10 | 62.32 |

The second experiment is training in parallel two ResNets divided into 2 modules on CIFAR100 [37]. We compare in Table 2 our results in this setup to the two delayed gradient methods DDG [32] and FR [31] from Table 2 in [31]. Here again, parallel TRGL has a better accuracy than the other two methods and than end-to-end training. With only two modules, the memory gains from less backpropagation are neutralized by the weight of the extra classifier and there are negligible memory savings compared to end-to-end training. However, parallel TRGL has a better test accuracy by up to almost 2 percentage points.

Table 2: Test accuracy of parallel TRGL with 2 modules (average and 95% confidence interval over 3 runs) on CIFAR100, compared to DDG, FR and E2E from Table 2 in [31].

| Architecture | Parallel VanGL | Parallel TRGL (ours) | DDG | FR | E2E |
|---|---|---|---|---|---|
| ResNet-101 | $77.31 \pm 0.27$ | $\mathbf{77.87} \pm 0.44$ | 75.75 | 76.90 | 76.52 |
| ResNet-152 | $75.40 \pm 0.75$ | $\mathbf{76.55} \pm 1.90$ | 73.61 | 76.39 | 74.80 |

The third experiment is training in parallel a ResNet-110 divided into two, four, eight and sixteen modules on STL10 [12]. We compare in Table 3 our results in this setup to the recent methods InfoPro [60] and DGL [8] from Table 2 in [60]. TRGL largely outperforms the other methods. It also outperforms end-to-end training in all but one case (that with 16 modules). With a batch size of 64, memory savings of parallel TRGL compared to end-to-end training reach $48\%$ and $58.5\%$ with 8 and 16 modules respectively, with comparable test accuracy. With 4 modules, TRGL training weighs $24\%$ less than end-to-end-training, and has a test accuracy that is better by 2 percentage points (see Section 4.2 and Table 6 for a detailed memory usage comparison with InfoPro).

Table 3: Test accuracy of Parallel (Par) TRGL with $K$ modules (average and 95% confidence interval over 5 runs) using a ResNet-110 on STL10, compared to DGL, two variants of InfoPro and E2E from Table 2 in [60].

| $K$ | Par VanGL | Par TRGL (ours) | DGL | InfoPro S | InfoPro C | E2E |
|---|---|---|---|---|---|---|
| 2 | $79.85 \pm 0.93$ | $\mathbf{80.04} \pm 0.85$ | $75.03 \pm 1.18$ | $78.98 \pm 0.51$ | $79.01 \pm 0.64$ | $77.73 \pm 1.61$ |
| 4 | $77.11 \pm 2.31$ | $\mathbf{79.72} \pm 0.81$ | $73.23 \pm 0.64$ | $78.72 \pm 0.27$ | $77.27 \pm 0.40$ | $77.73 \pm 1.61$ |
| 8 | $75.71 \pm 0.55$ | $\mathbf{77.82} \pm 0.73$ | $72.67 \pm 0.24$ | $76.40 \pm 0.49$ | $74.85 \pm 0.52$ | $77.73 \pm 1.61$ |
| 16 | $73.57 \pm 0.95$ | $\mathbf{77.22} \pm 1.20$ | $72.27 \pm 0.58$ | $73.95 \pm 0.71$ | $73.73 \pm 0.48$ | $77.73 \pm 1.61$ |

The fourth experiment is training (from scratch) in parallel a Swin-Tiny Transformer [39] divided into 4 modules on three datasets. We compare in Table 4 our results with those of InfoPro [60] and InfoProL, a variant of InfoPro proposed in [46]. TRGL outperforms the other module-wise training methods. It does not outperform end-to-end training in this case, but consumes $29\%$ less memory on CIFAR10 and CIFAR100 and $50\%$ less on STL10, compared to $38\%$ for InfoPro and $45\%$ for InfoProL in [46].

Table 4: Test accuracy of parallel TRGL with 4 modules (average and 95% confidence interval over 5 runs) on a Swin-Tiny Transformer, compared to InfoPro, InfoProL and E2E from Table 3 in [46], with memory saved compared to E2E as a percentage of E2E memory consumption in red.

| Dataset | Parallel VanGL | Parallel TRGL (ours) | InfoPro | InfoProL | E2E |
|---|---|---|---|---|---|
| STL10 | $67.00 \pm 1.36$ ($\downarrow 55\%$) | $\mathbf{67.92} \pm 1.12$ ($\downarrow 50\%$) | $64.61$ ($\downarrow 38\%$) | $66.89$ ($\downarrow 45\%$) | 72.19 |
| CIFAR10 | $83.94 \pm 0.42$ ($\downarrow 33\%$) | $\mathbf{86.48} \pm 0.54$ ($\downarrow 29\%$) | $83.38$ ($\downarrow 38\%$) | $86.28$ ($\downarrow 45\%$) | 91.37 |
| CIFAR100 | $69.34 \pm 0.91$ ($\downarrow 33\%$) | $\mathbf{74.11} \pm 0.31$ ($\downarrow 29\%$) | $68.36$ ($\downarrow 38\%$) | $73.00$ ($\downarrow 45\%$) | 75.03 |

Finally, we compare our method to InfoPro, DGL and Sedona in Table 5 below on a large scale experiment on ImageNet.

Table 5: Top 1 test accuracy of parallel TRGL with 2 modules on a ResNet-101 trained on ImageNet, compared to VanGL (baseline vanilla module-wise training), DGL and Sedona from [47] and InfoPro from [60] and end-to-end training.

| Dataset | Parallel VanGL | Parallel TRGL (ours) | DGL | Sedona | InfoPro | E2E |
|---|---|---|---|---|---|---|
| ImageNet | 78.11 | $\mathbf{79.41}$ | 78.47 | 79.28 | 78.15 | 78.71 |

## 4.2 Memory savings and training time

As seen above, parallel TRGL is lighter than end-to-end training by up to almost $60\%$. The extra memory consumed by our regularization compared to parallel VanGL is between 2 and $13\%$ of

end-to-end memory. Memory savings depend then mainly on the size of the auxiliary classifier, which can easily be adjusted. Note that delayed gradients method DDG and FR increase memory usage [31], and Sedona does not claim to save memory, but rather to speed up training [47]. DGL is architecture-wise essentially identical to VanGL and consumes the same memory.

We compare in Table 6 the memory consumption of our method to that of InfoPro [60] on a ResNet-110 on STL10 with a batch size of 64 (so the same setting as in Table 3). InfoPro [60] also propose to split the network into modules that have the same weight but not necessarily the same number of layers. They only implement this for $K \leq 4$ modules. When the modules are even in weight and not in depth, we call the training methods VanGL*, TRGL* and InfoPro*. In practice, this leads to shallower early modules which slightly hurts performance according to [47], and as seen below. However, TRGL* still outperforms InfoPro and end-to-end training, and it leads to even bigger memory savings than InfoPro*. We see in Table 6 below that TRGL saves more memory than InfoPro in two out of three cases (4 and 8 modules), and about the same in the third case (16 modules), with much better test accuracy in all cases. Likewise, TRGL* is lighter than InfoPro*, with better accuracy.

Table 6: Memory savings using a ResNet-110 on STL10 split into $K$ modules trained in parallel with a batch size of 64, as a percentage of the weight of end-to-end training. Average test accuracy over 5 runs is between brackets. Test accuracy of end-to-end training is 77.73%.

| $K$ | Equally deep modules | | | Equally heavy modules | | |
|---|---|---|---|---|---|---|
| | Par VanGL | Par TRGL (ours) | InfoPro | Par VanGL* | Par TRGL* (ours) | InfoPro* |
| 4 | 27% (77.11) | 24% (**79.72**) | 18% (78.72) | 41% (77.14) | 39% (**78.94**) | 33% (78.78) |
| 8 | 50% (75.71) | 48% (**77.82**) | 37% (76.40) | | | |
| 16 | 61% (73.57) | 58% (**77.22**) | 59% (73.95) | | | |

However, parallel module-wise training does slightly slow down training. Epoch time increases by 6% with 2 modules and by 16% with 16 modules. TRGL is only slower than VanGL by 2% for all number of modules due to the additional regularization term. This is comparable to InfoPro which reports a time overhead between 1 and 27% compared to end-to-end training. See Appendix F for details.

## 4.3 Sequential full block-wise training

Block-wise sequential training, meaning that each module is a single residual block and that the blocks are trained sequentially, therefore requiring only enough memory to train one block and its classifier. Even though it has been less explored in recent module-wise training methods, it has been used in practice in very constrained settings such as on-device training [58, 57]. We therefore test our regularization in this section in this setting, with more details in Appendix G.

We propose here to use shallower ResNets that are initially wider. These architectures are well-adapted to layer-wise training as seen in [7]. We check first in Table 10 in Appendix G that this architecture works well with parallel module-wise training with 2 modules by comparing it favorably on CIFAR10 [37] with methods DGL [8], InfoPro [60] and DDG [32] that use a ResNet-110 with the same number of parameters.

We then train a 10-block ResNet block-wise on CIFAR100. In Tables 11 and 12 in Appendix G, we see that MLS training improves the accuracy of sequential training by 0.8 percentage points when the trainset is full, but works less well on small train sets. Of the two, the regularization mainly improves the test accuracy of MLS training. The improvement increases as the training set gets smaller and reaches 1 percentage point. While parallel module-wise training performs quite close to end-to-end training in the full data regime and much better in the small data regime, sequential and multi-lap sequential training are competitive with end-to-end training in the small data regime. Combining the multi-lap trick and the regularization improves the accuracy of sequential training by 1.2 percentage points when using the entire trainset. We report further results for full block-wise training on MNIST [38] and CIFAR10 [37] in Tables 13 and 14 in Appendix G.

The 88% accuracy of sequential training on CIFAR10 in Table 13 is the same as in Table 2 of [7], which is the best method for layer-wise sequential training available, with VGG networks of comparable depth and width.

## 4.4 Accuracy after each module

Finally, we verify that our method avoids the stagnation or collapse in accuracy with depth. In Figure 2 below, we show the accuracy of each module with and without the regularization.

On the left, from parallel module-wise training experiments from Table 3, TRGL performs worse than vanilla greedy learning early, but surpasses it in later modules, indicating that it does avoid early overfitting. On the right, from sequential block-wise training experiments from Table 13, we see a large decline in performance that the regularization avoids. We see similar patterns in Figure 3 in Appendix G with parallel and MLS block-wise training.

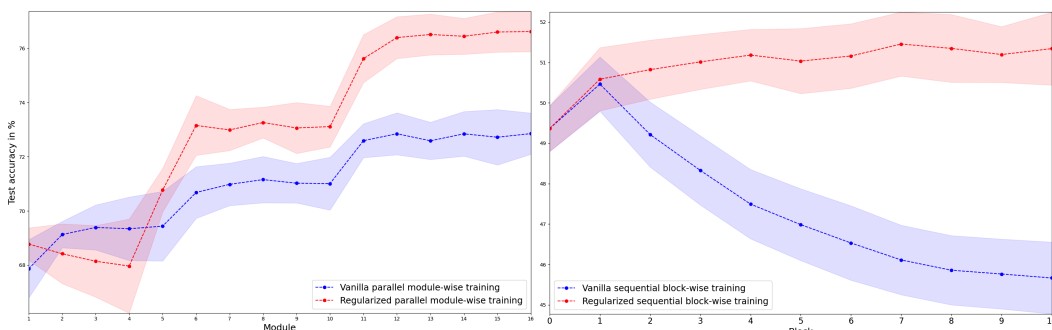

Figure 2: Test accuracy after each module averaged over 10 runs with 95% confidence intervals. Left: parallel vanilla (VanGL, in blue) and regularized (TRGL, in red) module-wise training of a ResNet-110 with 16 modules on STL10 (Table 3). Right: sequential vanilla (VanGL, in blue) and regularized (TRGL, in red) block-wise training of a 10-block ResNet on 2% of CIFAR10 (Table 13).

## 5   Limitations

The results in Appendix G show a few limitations of our method, as the improvements from the regularization are sometimes minimal on sequential training. However, the results show that our approach works in all settings (parallel and sequential with many or few modules), whereas other papers don't test their methods in all settings, and some show problems in other settings than the original one in subsequent papers (e.g. delayed gradients methods when the number of modules increases [31] and PredSim in [47]). Also, for parallel training in Section 4.1, the improvement from the regularization compared to VanGL is larger and increases with the number of modules (so with the memory savings) and reaches almost 5 percentage points. We show in Appendix H that our method is not very sensitive to the choice of hyperparameter $\tau$ over a large scale.

## 6   Related work

Layer-wise training was initially considered as a pre-training and initialization method [10, 43] and was recently shown to be competitive with end-to-end training [7, 45]. Many papers consider using a different auxiliary loss, instead of or in addition to the classification loss: kernel similarity [42], information-theory-inspired losses [53, 44, 41, 60] and biologically plausible losses [53, 45, 25, 11]. Methods [7], PredSim [45], DGL [8], Sedona [47] and InfoPro [60] report the best module-wise training results. [7, 8] do it simply through the architecture choice of the auxiliary networks. Sedona applies architecture search to decide on where to split the network into modules and what auxiliary classifier to use before module-wise training. Only BoostResNet [30] also proposes a block-wise training idea geared for ResNets. However, their results only show better early performance and end-to-end fine-tuning is required to be competitive. A method called ResIST [19] that is similar to block-wise training of ResNets randomly assigns ResBlocks to one of up to 16 modules that

are trained independently and reassembled before another random partition. More of a distributed training method, it is only compared to local SGD [54]. These methods can all be combined with our regularization, and we do use the auxiliary classifier from [7, 8].

Besides module-wise training, methods such as DNI [33, 14], DDG [32] and FR [31], solve the update and backward locking problems with an eye towards parallelization by using delayed or predicted gradients, or even predicted inputs to address forward locking, which is what [55] do. But they observe training issues with more than 5 modules [31]. This makes them compare unfavorably to module-wise training [8]. The high dimension of the predicted gradient, which scales with the size of the network, makes [33, 14] challenging in practice. Therefore, despite its simplicity, greedy module-wise training is more appealing when working in a constrained setting.

Viewing ResNets as dynamic transport systems [16, 36] followed from their view as a discretization of ODEs [61]. Transport regularization of ResNets in particular is motivated by the observation that they are naturally biased towards minimally modifying their input [35, 26]. We further linked this transport viewpoint with gradient flows in the Wasserstein space to apply it in a principled way to module-wise training. Gradient flows on the data distribution appeared recently in deep learning. In [1], the focus is on functionals of measures whose first variations are known in closed form and used, through their gradients, in the algorithm. This limits the scope of their applications to transfer learning and similar tasks. Likewise, [21, 40, 4, 3] use the explicit gradient flow of $f$-divergences and other distances between measures for generation and generator refinement. In contrast, we use the discrete minimizing movement scheme which does not require computation of the first variation and allows to consider classification.

## 7   Conclusion

We introduced a transport regularization for module-wise training that theoretically links module-wise training to gradient flows of the loss in probability space. Our method provably leads to more regular modules and experimentally improves the test accuracy of module-wise parallel, sequential and multi-lap sequential (a variant of sequential training that we introduce) training. Through this simple method that does not complexify the architecture, we make module-wise training competitive with end-to-end training while benefiting from its lower memory usage. Being a regularization, the method can easily be combined with other layer-wise training methods. Future work can experiment with working in Wasserstein space $W_p$ for $p \neq 2$, i.e. regularizing with a norm $\|.\|_p$ with $p \neq 2$.

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

## A  Background on optimal transport

The Wasserstein space $\mathbb{W}_2(\Omega)$ with $\Omega$ a convex and compact subset of $\mathbb{R}^d$ is the space $\mathcal{P}(\Omega)$ of probability measures over $\Omega$, equipped with the distance $W_2$ given by the solution to the optimal transport problem

$$W_2^2(\alpha, \beta) = \min_{\gamma \in \Pi(\alpha, \beta)} \int_{\Omega \times \Omega} \|x - y\|^2 \, d\gamma(x, y) \tag{12}$$

where $\Pi(\alpha, \beta)$ is the set of probability distribution over $\Omega \times \Omega$ with first marginal $\alpha$ and second marginal $\beta$, i.e. $\Pi(\alpha, \beta) = \{\gamma \in \mathcal{P}(\Omega \times \Omega) \mid \pi_{1\sharp}\gamma = \alpha, \ \pi_{2\sharp}\gamma = \beta\}$ where $\pi_1(x, y) = x$ and $\pi_2(x, y) = y$. The optimal transport problem can be seen as looking for a transportation plan minimizing the cost of displacing some distribution of mass from one configuration to another. This problem indeed has a solution in our setting and $W_2$ can be shown to be a geodesic distance (see for example [49, 59]). If $\alpha$ is absolutely continuous and $\partial\Omega$ is $\alpha$-negligible then the problem in (12) (called the Kantorovich problem) has a unique solution and is equivalent to the Monge problem, i.e.

$$W_2^2(\alpha, \beta) = \min_{T \text{ s.t. } T_\sharp \alpha = \beta} \int_\Omega \|T(x) - x\|^2 \, d\alpha(x) \tag{13}$$

and this problem has a unique solution $T^\star$ linked to the solution $\gamma^\star$ of (12) through $\gamma^\star = (\mathrm{id}, T^\star)_\sharp \alpha$. Another equivalent formulation of the optimal transport problem in this setting is the dynamical formulation [9]. Here, instead of directly pushing samples of $\alpha$ to $\beta$ using $T$, we can equivalently displace mass, according to a continuous flow with velocity $v_t : \mathbb{R}^d \to \mathbb{R}^d$. This implies that the density $\alpha_t$ at time $t$ satisfies the *continuity equation* $\partial_t \alpha_t + \nabla \cdot (\alpha_t v_t) = 0$, assuming that initial and final conditions are given by $\alpha_0 = \alpha$ and $\alpha_1 = \beta$ respectively. In this case, the optimal displacement is the one that minimizes the total action caused by $v$ :

$$W_2^2(\alpha, \beta) = \min_v \int_0^1 \|v_t\|_{L^2(\alpha_t)}^2 \, dt \tag{14}$$
$$\text{s.t. } \partial_t \alpha_t + \nabla \cdot (\alpha_t v_t) = 0, \ \alpha_0 = \alpha, \alpha_1 = \beta$$

Instead of describing the density's evolution through the continuity equation, we can describe the paths $\phi_t^x$ taken by particles at position $x$ from $\alpha$ when displaced along the flow $v$. Here $\phi_t^x$ is the position at time $t$ of the particle that was at $x \sim \alpha$ at time 0. The continuity equation is then equivalent to $\partial_t \phi_t^x = v_t(\phi_t^x)$. See chapters 4 and 5 of [49] for details. Rewriting the conditions as necessary, Problem (14) becomes

$$W_2^2(\alpha, \beta) = \min_v \int_0^1 \|v_t\|_{L^2((\phi_t^\cdot)_\sharp \alpha)}^2 \, dt \tag{15}$$
$$\text{s.t. } \partial_t \phi_t^x = v_t(\phi_t^x), \ \phi_0^\cdot = \mathrm{id}, (\phi_1^\cdot)_\sharp \alpha = \beta$$

and the optimal transport map $T^\star$ that solves (13) is in fact $T^\star(x) = \phi_1^x$ for $\phi$ that solves the continuity equation together with the optimal $v^\star$ from (15). We refer to [49, 59] for these results on optimal transport.

Optimal transport maps have some regularity properties under some boundedness assumptions. We mention the following result from [20]:

**Theorem A.1.** *Let $\alpha$ and $\beta$ be absolutely continuous measures on $\mathbb{R}^d$ and $T$ the optimal transport map between $\alpha$ and $\beta$ for the Euclidean cost. Suppose there are bounded open sets $X$ and $Y$, such that the density of $\alpha$ (respectively of $\beta$) is null on $X^{\mathsf{c}}$ (respectively $Y^{\mathsf{c}}$) and bounded away from zero and infinity on $X$ (respectively $Y$).*

*Then there exists two relatively closed sets of null measure $A \subset X$ and $B \subset Y$, such that $T$ is $\eta$-Hölder continuous from $X \setminus A$ to $Y \setminus B$, i.e. $\forall \, x, y \in X \setminus A$ we have*
$$\|T(x) - T(y)\| \le C\|x - y\|^\eta \text{ for constants } \eta, C > 0$$

## B  Background on gradient flows

We follow [50, 2] for this background on gradient flows. Given a function $\mathcal{L} : \mathbb{R}^d \to \mathbb{R}$ and an initial point $x_0 \in \mathbb{R}^d$, a *gradient flow* is a curve $x : [0, \infty[ \to \mathbb{R}^d$ that solves the Cauchy problem

$$\begin{cases} x'(t) = -\nabla \mathcal{L}(x(t)) \\ x(0) = x_0 \end{cases} \tag{16}$$

A solution exists and is unique if $\nabla \mathcal{L}$ is Lipschitz or $\mathcal{L}$ is convex. Given $\tau > 0$ and $x_0^\tau = x_0$ define a sequence $(x_k^\tau)_k$ through the *minimizing movement scheme*:

$$x_{k+1}^\tau \in \arg\min_{x \in \mathbb{R}^d} \; \mathcal{L}(x) + \frac{1}{2\tau}\|x - x_k^\tau\|^2 \tag{17}$$

$\mathcal{L}$ lower semi-continous and $\mathcal{L}(x) \geq C_1 - C_2\|x\|^2$ guarantees existence of a solution of (17) for $\tau$ small enough. $\mathcal{L}$ $\lambda$-convex meets these conditions and also provides uniqueness of the solution because of strict convexity of the target. See [49, 50, 2].

We interpret the point $x_k^\tau$ as the value of a curve $x$ at time $k\tau$. We can then construct a curve $x^\tau$ as the piecewise constant interpolation of the points $x_k^\tau$. We can also construct a curve $\tilde{x}^\tau$ as the affine interpolation of the points $x_k^\tau$.

If $\mathcal{L}(x_0) < \infty$ and $\inf \mathcal{L} > -\infty$ then $(x^\tau)$ and $(\tilde{x}^\tau)$ converge uniformly to the same curve $x$ as $\tau$ goes to zero (up to extracting a subsequence). If $\mathcal{L}$ is $\mathcal{C}^1$, then the limit curve $x$ is a solution of (16) (i.e. a gradient flow of $\mathcal{L}$). If $\mathcal{L}$ is not differentiable then $x$ is solution of the problem defined using the subdifferential of $\mathcal{L}$, i.e. $x$ satisfies $x'(t) \in -\partial \mathcal{L}(x(t))$ for almost every $t$.

If $\mathcal{L}$ is $\lambda$-convex with $\lambda > 0$, then the solution to (16) converges exponentially to the unique minimizer of $\mathcal{L}$ (which exists by coercivity). So taking $\tau \to 0$ and $k \to \infty$, we tend towards the minimizer of $\mathcal{L}$.

The advantage of the minimizing movement scheme (17) is that it can be adapted to metric spaces by replacing the Euclidean distance by the metric space's distance. In the (geodesic) metric space $\mathbb{W}_2(\Omega)$ with $\Omega$ convex and compact, for $\mathcal{L} : \mathbb{W}_2(\Omega) \to \mathbb{R} \cup \{\infty\}$ lower semi-continuous for the weak convergence of measures in duality with $\mathcal{C}(\Omega)$ (equivalent to lower semi-continuous with respect to the distance $W_2$) and $\rho_0^\tau = \rho_0 \in \mathcal{P}(\Omega)$, the minimizing movement scheme (17) becomes

$$\rho_{k+1}^\tau \in \arg\min_{\rho \in \mathcal{P}(\Omega)} \; \mathcal{L}(\rho) + \frac{1}{2\tau}W_2^2(\rho, \rho_k^\tau) \tag{18}$$

This problem has a solution because the objective is lower semi-continuous and the minimization is over $\mathcal{P}(\Omega)$ which is compact by Banach-Alaoglu.

We can construct a piecewise constant interpolation between the measures $\rho_k^\tau$, or a geodesic interpolation where we travel along a geodesic between $\rho_k^\tau$ and $\rho_{k+1}^\tau$ in $\mathbb{W}_2(\Omega)$, constructed using the optimal transport map between these measures. Again, if $\mathcal{L}(x_0) < \infty$ and $\inf \mathcal{L} > -\infty$ then both interpolations converge uniformly to a limit curve $\tilde{\rho}$ as $\tau$ goes to zero. Under further conditions on $\mathcal{L}$, mainly $\lambda$-geodesic convexity (i.e. $\lambda$-convexity along geodesics) for $\lambda > 0$, we can prove stability and convergence of $\tilde{\rho}(t)$ to a minimizer of $\mathcal{L}$ as $t \to \infty$, see [49, 50, 2].

## C  Proof of Proposition 2.3

*Proof.* Take a minimizing sequence $(\tilde{F}_i, \tilde{T}_i)$, i.e. such that $\mathcal{C}(\tilde{F}_i, \tilde{T}_i) \to \min \mathcal{C}$, where $\mathcal{C} \geq 0$ is the target function in (3) and denote $\beta_i = \tilde{T}_{i\sharp}\rho_k^\tau$. Then by compacity $\tilde{F}_i \to F^\star$ and $\beta_i \rightharpoonup \beta^\star$ in duality with $\mathcal{C}_b(\Omega)$ by Banach-Alaoglu. There exists $T^\star$ an optimal transport map between $\rho_k^\tau$ and $\beta^\star$. Then $\mathcal{C}(F^\star, T^\star) \leq \lim \mathcal{C}(\tilde{F}_i, \tilde{T}_i) = \min \mathcal{C}$ by continuity of $\mathcal{L}$ and because

$$\int_\Omega \|T^\star(x) - x\|^2 \, d\rho_k^\tau(x) = W_2^2(\rho_k^\tau, \beta^\star)$$

$$= \lim W_2^2(\rho_k^\tau, \beta_i)$$

$$\leq \lim \int_\Omega \|\tilde{T}_i(x) - x\|^2 \, d\rho_k^\tau(x)$$

as $W_2$ metrizes weak convergence of measures. We take $(F_k^\tau, T_k^\tau) = (F^\star, T^\star)$. It is also immediate that for any minimizing pair, the transport map has to be optimal. Taking a minimizing sequence $(\tilde{F}_i, \tilde{v}^i)$ and the corresponding induced maps $\tilde{T}_i$ we get the same result for Problem (11). Problems (3) and (11) are equivalent by the equivalence between Problems (13) and (15). □

## D  Varying the regularization weight

The discussion in Section 2.2 suggests taking a fixed weight $\tau$ for the transport cost that is as small as possible. However, instead of using a fixed $\tau$, we might want to vary it along the depth $k$ to further

constrain with a smaller $\tau_k$ the earlier modules to avoid that they overfit or the later modules to maintain the accuracy of earlier modules. We might also want to regularize the network further in earlier epochs when the data is more entangled. To unify and formalize this varying weight $\tau_{k,i}$ across modules $k$ and SGD iterations $i$, we use a scheme inspired by the method of multipliers to solve Problems (2) and (10). To simplify the notations, we will instead consider the weight $\lambda_{k,i} := 2\tau_{k,i}$ given to the loss. We denote $\theta_{k,i}$ the parameters of both $T_k$ and $F_k$ at SGD iteration $i$. We also denote $L(\theta, x)$ and $W(\theta, x)$ respectively the loss and the transport regularization as functions of parameters $\theta$ and data $x$. We now increase the weight $\lambda_{k,i}$ of the loss every $s$ iterations of SGD by a value that is proportional to the current loss. Given increase factor $h > 0$, initial parameters $\theta_{k,1}$, initial weights $\lambda_{k,1} \geq 0$, learning rates $(\eta_i)$ and batches $(x_i)$, we apply for module $k$ and $i \geq 1$:

$$\begin{cases} \theta_{k,i+1} & = \theta_{k,i} - \eta_i \nabla_\theta(\lambda_{k,i} \ L(\theta_{k,i}, x_i) + W(\theta_{k,i}, x_i)) \\ \lambda_{k,i+1} & = \lambda_{k,i} + hL(\theta_{k,i+1}, x_{i+1}) \text{ if } i \bmod s = 0 \text{ else } \lambda_{k,i} \end{cases}$$

The weights $\lambda_{k,i}$ will vary along modules $k$ because they will evolve differently with iterations $i$ for each $k$. They will increase more slowly with $i$ for larger $k$ because deeper modules will have smaller loss. This method can be seen as a method of multipliers for the problem of minimizing the transport under the constraint of zero loss. Therefore it is immediate by slightly adapting the proof of Proposition 2.3 or from [36] that we are still solving a problem that admits a solution whose non-auxiliary part is an optimal transport map with the same regularity as stated above. We use the same initial value $\lambda_1 = \lambda_{k,1}$ for all modules so that this method requires choosing three hyper-parameters ($h$, $s$ and $\lambda_1$). In practice (see Section 4.1 and Appendix E), it works best in only one experiment. Simply manually finding a value of $\tau$ that is multiplied by 2 for the second half of the network works best in all the other experiments.

## E  Implementation details

We use standard data augmentation and standard implementations for VGG-19, ResNet-50, ResNet-101, ResNet-110, ResNet-152 and Swin-Tiny Transformer (the same as for the other methods in Section 4.1). We use NVIDIA Tesla V100 16GB GPUs for the experiments. Training a Resnet-152 on TinyImageNet in Table 1 takes about 36 hours. Training a Resnet-152 on CIFAR100 in Table 1 takes about 11 hours. Training a ResNet-110 on STL10 in Table 3 takes about 3 hours. Training a Swin-Tiny Transformer in Table 4 take between 40 minutes and 1 hour.

For sequential and multi-lap sequential training, we use SGD with a learning rate of 0.007. With the exception of the Swin Transformer in Table 4, we use SGD for parallel training with learning rate of 0.003 in all Tables but Table 3 where the learning rate is 0.002. For the Swin Transformer in Table 4, we use the AdamW optimizer with a learning rate of 0.007 and a CosineLR scheduler.

For end-to-end training we use a learning rate of 0.1 that is divided by five at epochs 120, 160 and 200. Momentum is always 0.9. For parallel and end-to-end training, we train for 300 epochs. For sequential and multi-lap sequential training, the number of epochs varies per module (see Section G).

For experiments in Section 4.1, we use a batch size of 256, orthogonal initialization [51] with a gain of 0.1, label smoothing of 0.1 and weight decay of 0.0002. The batch size changes to 64 for Table 3 and to 1024 for Table 4.

For experiments in Section 4.3, we use a batch size of 128, orthogonal initialization with a gain of 0.05, no label smoothing and weight decay of 0.0001.

In Table 1, we use $\tau = 500000$ for the first two modules and then double it for the last two modules for TRGL. In Table 2, we use $\lambda_{k,1} = 1, h = 1$ and $s = 50$ for TRGL. In Table 3, we use $\tau = 0.5$ and double it at the midpoint, expect for the first line where $\tau = 50$.

## F  Memory savings and training time

We compare in Table 7 the memory consumption of our method to that of InfoPro [60] on a ResNet-110 split into $K$ modules trained in parallel on STL10 with a batch size of 64 (so the same setting as in Table 3 in Section 4.1). We report in Table 7 the memory saved as a percentage of the 6230 MiB memory required by end-to-end training with the same batch size. VanGL refers to our architecture trained without the regularization. InfoPro [60] also propose to split the network into $K$ modules that

have the same weight but not necessarily the same number of layers. They only implement this for $K \leq 4$ modules. When the modules are even in weight and not in depth, we call the training methods VanGL*, TRGL* and InfoPro*. In practice, this leads to shallower early modules which slightly hurts performance according to [47]. We verify this in Table 8 (to be compared with Table 3 in Section 4.1). However, TRGL* still outperforms InfoPro and end-to-end training, and it leads to even bigger memory savings. We see in Table 7 that TRGL saves more memory than InfoPro in two out of three cases (4 and 8 modules), and about the same in the third case (16 modules), with much better test accuracy in all cases. Likewise, TRGL* is lighter than InfoPro*, with better accuracy. We also see that the added memory cost of the regularization compared to vanilla greedy learning is small. However, parallel module-wise training does slightly slow down training. Epoch time increases by $6\%$ with 2 modules and by $16\%$ with 16 modules. TRGL is only slower than VanGL by $2\%$ for for all number of modules due to the additional regularization term. This is comparable to InfoPro which report a time overhead between 1 and $27\%$ compared to end-to-end training.

Table 7: Memory savings using a ResNet-110 on STL10 split into $K$ modules trained in parallel with a batch size of 64, as a percentage of the weight of end-to-end training. Average test accuracy over 5 runs is between brackets. Test accuracy of end-to-end training is $77.73\%$.

| | Equally deep modules | | | Equally heavy modules | | |
|---|---|---|---|---|---|---|
| $K$ | Par VanGL | Par TRGL (ours) | InfoPro | Par VanGL* | Par TRGL* (ours) | InfoPro* |
| 4 | 27% (77.11) | 24% (**79.72**) | 18% (78.72) | 41% (77.14) | 39% (**78.94**) | 33% (78.78) |
| 8 | 50% (75.71) | 48% (**77.82**) | 37% (76.40) | | | |
| 16 | 61% (73.57) | 58% (**77.22**) | 59% (73.95) | | | |

Table 8: Test accuracy of parallel (Par) TRGL* with $K$ modules (average and $95\%$ confidence interval over 5 runs) on a ResNet-110 trained on STL10, compared to InfoPro* and E2E training from Table 3 in [60]

| $K$ | Par VanGL* | Par TRGL* (ours) | InfoPro* |
|---|---|---|---|
| 2 | $79.05 \pm 1.33$ | $\mathbf{79.47} \pm 1.36$ | $79.05 \pm 0.57$ |
| 4 | $77.14 \pm 1.23$ | $\mathbf{78.94} \pm 1.13$ | $78.78 \pm 0.72$ |

Note that methods DDG [32] and FR [31], being delayed gradient methods and not module-wise training methods, do no save memory (they actually increase memory usage, see FR [31]). Sedona [47] also does not claim to save memory, as their first module (the heaviest) is deeper than the others, but rather to speed up computation. Finally, DGL [8] is architecture-wise essentially identical to VanGL and consumes the same memory.

## G   Sequential full block-wise training

To show that our method works well with all types of module-wise training when using few modules, we train a ResNet-101 split in 2 modules on CIFAR100, sequentially and multi-lap sequentially. Results are in Table 9. We see that our idea of multi-lap sequential training adds one percentage point of accuracy to sequential training, and that the regularization further improves the accuracy by about half a percentage point. As only one module has to be trained at a time, these two training methods require only around half the memory end-to-end training requires (the size of the heaviest module and its classifier more exactly).

Table 9: Test accuracy of sequential (Seq) and multi-lap sequential (MLS) TRGL and VanGL with 2 modules on CIFAR100 using ResNet-101 (average of 2 runs).

| Seq VanGL | Seq TRGL | MLS VanGL | MLS TRGL |
|---|---|---|---|
| 73.31 | 73.61 | 74.34 | **74.78** |

We now focus on full block-wise training, meaning that each module is a single ResBlock, mostly sequentially. We propose here to use shallower and initially wider ResNets with a downsampling and 256 filters initially and a further downsampling and doubling of the number of filters at the midpoint, no matter the depth. In these ResNets, we use the ResBlock from [27] with two convolutional layers. If such a network is divided in $K$ modules of $M$ ResBlocks each, we call the network a $K-M$ ResNet. These wider shallower architectures are well-adapted to layer-wise training as seen in [7]. We check in Table 10 that this architecture works well with parallel module-wise training by comparing favorably on CIFAR10 ([37]) a 2-7 ResNet with DGL, InfoPro ([60]) and DDG [32]. The 2-7 ResNet has 45 millions parameters, which is about the same as the ResNet-110 divided in two used by the other methods, and performs better when trained in parallel.

Table 10: Average test accuracy and 95% confidence interval of 2-7 ResNet over 10 runs on CIFAR10 with parallel TRGL and VanGL, compared to DGL and DDG from [8] and InfoPro from [60] that split a ResNet-110 in 2 module-wise-parallel-trained modules.

| Parallel VanGL (ours) | Parallel TRGL (ours) | DGL | DDG | InfoPro |
|---|---|---|---|---|
| 94.01 ± .17 | **94.05** ± .18 | 93.50 | 93.41 | 93.58 |

We now train a 10-block ResNet block-wise on CIFAR100 (a 10-1 ResNet in our notations). We report even the small improvements in accuracy to show that our method works in all settings (parallel or sequential with many or few splits), which other methods don't do. For sequential training, block $k$ is trained for $50+10k$ epochs where $0 \le k \le 10$, block 0 being the encoder. This idea of increasing the number of epochs along with the depth is found in [43]. For MLS training, block $k$ is trained for $10+2k$ epochs, and this is repeated for 5 laps. In block-wise training, the last block does not always perform the best and we report the accuracy of the best block. In Table 11, we see that MLS training improves the test accuracy of sequential training by around $0.8$ percentage points when the training dataset is full, but works less well on small training sets. Of the two, the regularization mainly improves the test accuracy of MLS training. The improvement increases as the training set gets smaller and reaches 1 percentage point. That is also the case for parallel module-wise training in Table 12, which already performs quite close to end-to-end training in the full data regime and much better in the small data regime. Combining the multi-lap trick and the regularization improves the performance of sequential training by 1.2 percentage points.

Table 11: Average highest test accuracy and 95% confidence interval of 10-1 ResNet over 10 runs on CIFAR100 with different train sizes and sequential (Seq), multi-lap sequential (MLS) and parallel (Par) TRGL and VanGL, compared to E2E.

| Train size | Seq VanGL | Seq TRGL | MLS VanGL | MLS TRGL | E2E |
|---|---|---|---|---|---|
| 50000 | 68.74 ± 0.45 | **68.79** ± 0.56 | 69.48 ± 0.53 | **69.95** ± 0.50 | 75.85 ± 0.70 |
| 25000 | 60.48 ± 0.15 | **60.59** ± 0.14 | 61.33 ± 0.23 | **61.71** ± 0.32 | 65.36 ± 0.31 |
| 12500 | 51.64 ± 0.33 | **51.74** ± 0.26 | 51.30 ± 0.22 | **51.89** ± 0.30 | 52.39 ± 0.97 |
| 5000 | 36.37 ± 0.33 | **36.40** ± 0.40 | 33.68 ± 0.48 | **34.61** ± 0.59 | 36.38 ± 0.31 |

Table 12: Average highest test accuracy and 95% confidence interval of 10-1 ResNet over 10 runs on CIFAR100 with different train sizes and sequential (Seq), multi-lap sequential (MLS) and parallel (Par) TRGL and VanGL, compared to E2E.

| Train size | Par VanGL | Par TRGL | E2E |
|---|---|---|---|
| 50000 | 72.59 ± 0.40 | **72.63** ± 0.40 | 75.85 ± 0.70 |
| 25000 | 64.84 ± 0.19 | **65.01** ± 0.27 | 65.36 ± 0.31 |
| 12500 | 55.13 ± 0.24 | **55.40** ± 0.35 | 52.39 ± 0.97 |
| 5000 | 39.45 ± 0.23 | **40.36** ± 0.23 | 36.38 ± 0.31 |

We report further results of block-wise training on CIFAR10 in Table 13 and on MNIST [38] in Table 14, but now we report the accuracy of the last block. We see again greater improvement due to the regularization as the training set gets smaller, gaining up to 6 percentage points.

Table 13: Average last block test accuracy and 95% confidence interval of 10-1 ResNet over 10 runs on CIFAR10 with different train sizes and sequential TRGL and VanGL, compared to E2E.

| Train size | Seq VanGL | Seq TRGL | E2E |
|---|---|---|---|
| 50000 | $88.02 \pm .18$ | $\mathbf{88.20} \pm .24$ | $91.88 \pm .18$ |
| 25000 | $83.95 \pm .13$ | $\mathbf{84.28} \pm .22$ | $88.75 \pm .27$ |
| 10000 | $76.00 \pm .39$ | $\mathbf{77.18} \pm .34$ | $82.61 \pm .35$ |
| 5000 | $67.74 \pm .49$ | $\mathbf{69.67} \pm .44$ | $73.93 \pm .67$ |
| 1000 | $45.67 \pm .88$ | $\mathbf{51.34} \pm .90$ | $50.63 \pm .98$ |

Table 14: Average last block test accuracy and 95% confidence interval of 20-1 ResNet (32 filters, fixed encoder, same classifier) over 20/50 runs on MNIST with different train sizes and parallel TRGL and VanGL, compared to E2E.

| Train size | Par VanGL | Par TRGL | E2E |
|---|---|---|---|
| 60000 | $99.07 \pm .04$ | $\mathbf{99.08} \pm .04$ | $99.30 \pm .03$ |
| 30000 | $98.90 \pm .05$ | $\mathbf{98.93} \pm .06$ | $99.22 \pm .03$ |
| 12000 | $98.52 \pm .06$ | $\mathbf{98.59} \pm .06$ | $98.96 \pm .06$ |
| 6000 | $98.05 \pm .09$ | $\mathbf{98.16} \pm .07$ | $98.62 \pm .06$ |
| 1500 | $96.34 \pm .12$ | $\mathbf{96.91} \pm .07$ | $97.19 \pm .08$ |
| 1200 | $95.80 \pm .12$ | $\mathbf{96.58} \pm .09$ | $96.88 \pm .09$ |
| 600 | $91.35 \pm .99$ | $\mathbf{95.16} \pm .15$ | $95.30 \pm .17$ |
| 300 | $89.81 \pm .73$ | $\mathbf{92.86} \pm .24$ | $92.87 \pm .28$ |
| 150 | $81.84 \pm 1.22$ | $\mathbf{87.48} \pm .42$ | $87.82 \pm .59$ |

The 88% accuracy of sequential training on CIFAR10 in Table 13 is the same as for sequential training in table 2 of [7], which is the best method for layer-wise sequential training available, with VGG networks of comparable depth and width.

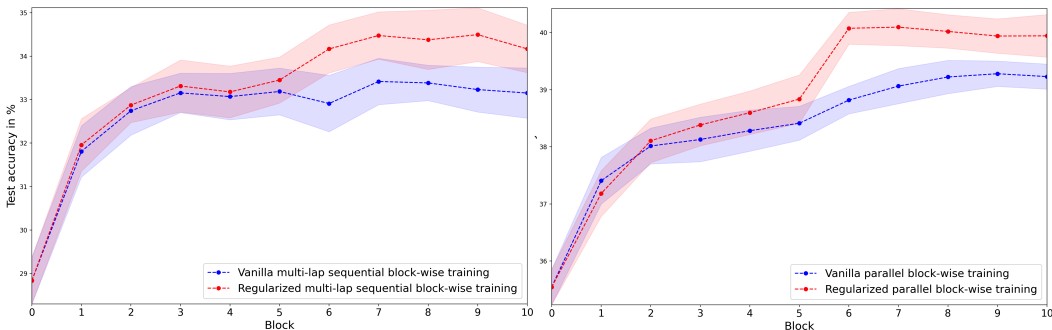

Figure 3: Test accuracy after each block of 10-1 ResNet averaged over 10 runs with 95% confidence intervals. Left: multi-lap sequential vanilla (VanGL, in blue) and regularized (TRGL, in red) block-wise training on 10% of the CIFAR100 training set. Right: parallel vanilla (VanGL, in blue) and regularized (TRGL, in red) block-wise training on 10% of CIFAR100 training set.

# H  Sensitivity to hyper-parameters

We show in Figure 4 below that TRGL still performs better than VanGL (in the same setting as in Table 3 in Section 4.1) for values of $\tau$ from 0.03 to 100 and is still roughly equivalent to it for values up to 5000.

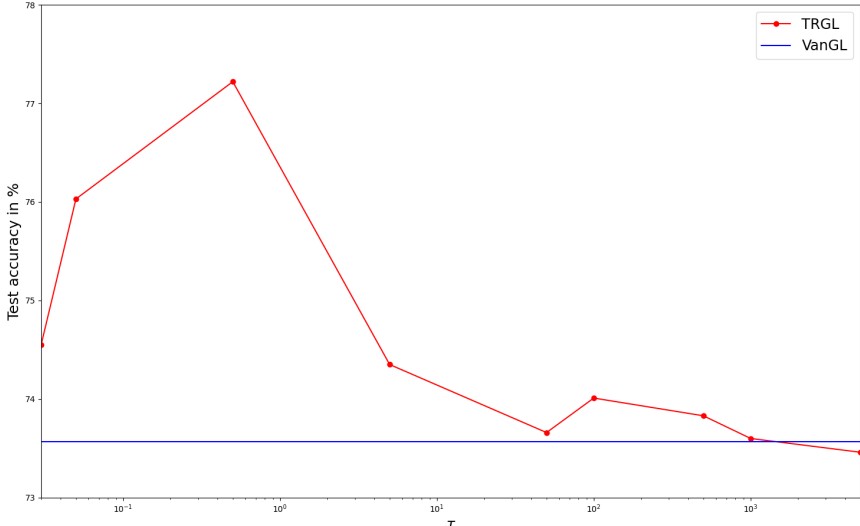

Figure 4: Average test accuracy over 5 runs of parallel TRGL using a ResNet110 on STL10 with 16 modules with different values of $\tau$ (in red), and of VanGL (blue line).

# I  Broader impact

Less memory usage has a positive environmental impact and allows organizations with less resources to use deep learning.

