# OpenReview forum: "Module-wise Training of Neural Networks via the Minimizing Movement Scheme"
_NeurIPS.cc/2023/Conference — NeurIPS 2023 poster_

### Official Review · Reviewer_N1H7 · 2023-06-27

**Soundness:** 3 good
**Presentation:** 3 good
**Contribution:** 3 good
**Rating:** 5
**Confidence:** 4

**Summary:**

This paper proposes a new training method for greedy layer-wise or module-wise training of neural networks, which is compelling in constrained and on-device settings where memory is limited and suffers from a stagnation problem. Experimental results show that their method improves the accuracy of module-wise training of various architectures.

-------After Rebuttal-------

I thank the authors for the answers to my comments and questions. Most of my concerns are properly addressed. Therefore I raise my score.

While I still think the word "early overfitting" in the paper should be carefully used. I agree with the phenomenon that vanilla module-wise training performs very well in the early layers but stagnates and gets overtaken later. While it can be due to the insufficient mutual information between the learned features and the inputs, instead of the early layer overfitting. Generally, I think the early modules containing fewer learnable parameters are difficult to overfit the large-scale dataset, such as ImageNet.

**Strengths:**

1. Overall, the idea is new and the proposed method is theoretically sound.
2. The paper is well-written and easy to follow.

**Weaknesses:**

On Line 34, the authors claim that the early modules can overfit and learn more discriminative features than end-to-end training, destroying task-relevant information, and deeper modules don’t improve the test accuracy significantly, or even degrade it. However, the experimental results in this paper are not sufficient enough to support this assumption, especially for the overfitting problem.

As the listed experiments are all conducted on small or medium-scale datasets, the overfitting problem might happen for the early layers. While, for the large-scale dataset, such as ImageNet, the early layers are more likely to be under-fitting due to the insufficient model
capacity of the first few layers.

**Questions:**

1. The wall-time training cost is also an important evaluation metric for training algorithms. I suggest the authors provide a comparison of the training time between their proposed TRGL and other related methods.

**Limitations:**

See weaknesses and questions.

---

> ### Author Rebuttal · Authors · 2023-08-04
>
> Dear Reviewer N1H7,
>
> Thank you for your valuable review. We answer your remarks below.
>
> **Weaknesses**
>
> **1. Evidence for early overfitting**
>
> This is indeed an interesting question. Our experiments in Figures 2 and 3 demonstrate that vanilla module-wise training performs very well in the early layers but stagnates and gets overtaken later for the datasets considered in the paper.
>
> Most importantly, we cite 4 papers [36,5,53,40] that make the same observation on line 33. The arguments developed in each paper are detailed below. We therefore considered that this was well-established in the literature. We propose to add this discussion of the evidence in the literature for this early overfitting phenomenon to the paper.
>
> [36] compare the accuracy after each layer to that of end-to-end training and find that layer-wise training overfits and outperforms E2E in early layers, but stagnates later while E2E improves dramatically.
>
> [53] explore this in more detail. They find that "greedy SL (module-wise training) contributes to dramatically more discriminative features with the first one (or few) local module(s), but is only able to slightly improve the performance with all the consequent modules. In contrast, the E2E learned network progressively boosts the linear separability of features throughout the whole network with even more significant effects in the later layers, surpassing greedy SL eventually". [53] also show that module-wise training detroys a larger amount of mutual information between the learned features and the inputs (and between the learned features and the outputs) compared to end-to-end training in the early modules.
>
> Also, [40] find that the first module "may learn only too coarse features to classify complex images, which may be less meaningful representations for upper layers". They solve this by making the first module deeper, which reduces the memory savings of the method, and [5] solve the problem by making the auxiliary network deeper.
>
> **2. Large scale experiments**
>
> Since submitting the paper we have run experiments on ImageNet (ResNet-101 split into 2 modules trained in parallel). Results are below and in the additional PDF file. This validates that the conclusion drawn on the smaller datasets still holds for large datasets such as ImageNet. We did not experiment with more than 2 modules on ImageNet due to resource limitations.
>
> | Dataset  | Parallel VanGL | Parallel TRGL (ours) | DGL   | Sedona | InfoPro | E2E   |
> |----------|----------------|----------------------|-------|--------|---------|-------|
> | ImageNet | 78.11          | **79.41**            | 78.47 | 79.28  | 78.15   | 78.71 |
>
>
>
> **Questions**
>
> **1. Training time**
>
> This was included in the paper but probably not prominently enough, in section 4.2 starting line 259:
>
> Vanilla parallel module-wise (denoted VanGL in the paper) training does slightly increase  training time. Epoch time increases by 6% with 2 modules and by 16% with 16 modules compared to end-to-end training. By adding our regularization, TRGL is only slower than VanGL by 2% for any number of modules. This is comparable to InfoPro baseline which reports a time overhead between 1 and 27% compared to end-to-end training.
>
> Note that InfoPro is the most comparable method to ours in this regard as it also saves memory at the price of slowing down training a little bit, while having performances that are close to or better than end-to-end training.

---

> ### Author Response · Authors · 2023-08-18
> **After review update**
>
> Indeed the term 'destruction of information' as in [53] in our rebuttal is better than 'overfitting' to describe this phenomenon.

---

> ### Comment · Area_Chair_JDVT · 2023-08-21
> **Can you please check the rebuttal comments?**
>
> Dear reviewer,
>
> The authors have provided a response to your comments. Can you please take a look and accordingly comment, and updated your review?
>
> Thanks,
> -Area Chair

---

### Official Review · Reviewer_X5WU · 2023-06-29

**Soundness:** 3 good
**Presentation:** 3 good
**Contribution:** 3 good
**Rating:** 6
**Confidence:** 4

**Summary:**

The paper explores the module-wise training of neural networks via the Minimizing Movement Scheme. This approach aims to overcome the stagnation problem often encountered in layer-wise training, leading to improved accuracy and reduced memory usage. The authors compare their results with those of other methods and demonstrate the effectiveness of their approach on various architectures, including ResNets, Transformers, and VGG.

**Strengths:**

1. The introduced methodology effectively addresses the frequently observed stagnation issue associated with layer-wise training. As a result, there is a noteworthy enhancement in model accuracy along with a commendable reduction in memory usage.

2. Establishing a connection between neural network training and optimal transport theory is not only intellectually stimulating, but it also presents intriguing theoretical insights.

3. Generally, the manuscript is well-structured and clearly articulated. However, some convoluted notations used in Section 2.2 could potentially be simplified for improved reader comprehension.




**Weaknesses:**

1. The paper is currently missing a detailed convergence analysis. It's vital to elucidate whether the overall training process will indeed converge in order to reinforce the reliability and robustness of the proposed approach.

2. The study could benefit from a more comprehensive ablation analysis, specifically, an exploration on the impact of variations in the regularization penalty. This would further validate the robustness of the model and provide additional insights into its behavior under different conditions.

**Questions:**

N/A

**Limitations:**

Yes

---

> ### Author Rebuttal · Authors · 2023-08-05
>
> Dear Reviewer X5WU,
>
> Thank you for your valuable review. We answer your remarks below.
>
> **Weaknesses**
>
> **1. Convergence**
>
> Indeed this is an important question and this will be further discussed in the final version. In Section 2.2, we have assumed that the training of the individual modules converges.
>
> Indeed, the training of shallow networks is known to converge [A,B,C,D,E]. Papers [5,6,24,F,G] show the convergence of vanilla module-wise (both parallel and sequential) training by chaining existing convergence results for shallow modules. We have not considered the impact of the regularization on this, but it is a simple quadratic regularization that makes the target more convex and [29] show the beneficial effects of this transport regularization on training a network end-to-end.
>
> Assuming therefore that the training of each shallow module converges with the number of epochs, the assumptions under which we have convergence as the regularization weight parameter $\tau$ goes to zero and the number of modules $k$ increases are discussed in Section 2.2 (lines 130 to 136). Convergence as the number of modules increases is the central point in module-wise training as we precisely want the modules to build upon each other to solve the task, and this is exactly what we demonstrate.
>
> [A] *Understanding Deep Neural Networks with Rectified Linear Units*, Arora et al., ICLR 2018
>
> [B] *Breaking the Curse of Dimensionality with Convex Neural Networks*, Bach, JMLR 2017
>
> [C] *Beating the Perils of Non-Convexity: Guaranteed Training of Neural Networks using Tensor Methods*, Janzamin et al, arXiv 2016
>
> [D] *Learning One-hidden-layer Neural Networks with Landscape Design*, Ge et al., ICLR 2018
>
> [E] *Improved Learning of One-hidden-layer Convolutional Neural Networks with Overlaps*, Du et al., arXiv 2018
>
> [F] *A Provably Correct Algorithm for Deep Learning that Actually Works*, Malach et al., arXiv 2018
>
> [G] *Provable Bounds for Learning Some Deep Representations*, Arora et al., ICML 2014
>
>
> **2. Impact of variations in the regularization penalty**
>
> We already provide an exploration of the impact of variations in the regularization penalty in Figure 4 in Appendix H (this is mentioned in Section 5) on the STL10 dataset. We found that "TRGL performs better than VanGL for values of $\tau$ from 0.03 to 100 and is then roughly equivalent to it for values up to 5000."
>
> We have run a similar experiment using the Swin-Tiny Transformer split in 4 modules trained in parallel on CIFAR100 (line 4 in Table 4), with similar results. The figure is in the additional PDF joined to the general answer above.

---

> ### Comment · Area_Chair_JDVT · 2023-08-21
> **Can you please check the rebuttal comments?**
>
> Dear reviewer,
>
> The authors have provided a response to your comments. Can you please take a look and accordingly comment, and updated your review?
>
> Thanks,
> -Area Chair

---

### Official Review · Reviewer_U1e3 · 2023-07-06

**Soundness:** 2 fair
**Presentation:** 3 good
**Contribution:** 3 good
**Rating:** 6
**Confidence:** 3

**Summary:**

The paper proposed a new regularization for module-wise training via the distance of the input and output of the module.

**Strengths:**

1. The proposed regularization is quite straightforward and easy to apply.
2. The paper connects the proposed regularization with optimal transport via theoretical analysis.
3. Some experimental results are good, like adding more modules on STL-10.

**Weaknesses:**

1. Although the author shows the connection between the proposed regularization and optimal transport, it does not provide insights regarding convergence. Whether the proposed regularization helps the convergence of the multi-module training might be more interesting for larger audiences.
2. Authors argue that 'residual connections are already biased towards small displacements.' However, residual connections are not toward small values for early layers, as shown in [28]. As a result, how does the proposed method control each module's regularization strength? If they are set to the same value, then the regularization may be harmful to early layers. In addition, the selection of the regularization strength becomes more complex when increasing the number of modules K.
3. The improvements over the previous method are only obvious on STL-10, and STL-10 is not a large dataset. In addition, larger datasets and models are omitted. I think memory saving will be more meaningful for larger datasets and models. As a result, the scalability of the proposed method is not examined.

**Questions:**

1. Why is the memory saving of the proposed method often less than other methods? Where this overhead comes from?
2. Some notations are not introduced properly. For example, the `#' notation, and what is the difference of $T_k$ and $T_k^{\tau}$.

**Limitations:**

Limitations are provided.

---

> ### Author Rebuttal · Authors · 2023-08-04
>
> Dear Reviewer U1e3,
>
> Thank you for your valuable review, we answer your remarks below.
>
> **Weaknesses**
>
> **1. Convergence**
>
> Indeed this is an important question and this will be further discussed in the final version. In Section 2.2, we have assumed that the training of the individual modules converges.
>
> Indeed, the training of shallow networks is known to converge [A,B,C,D,E]. Papers [5,6,24,F,G] show the convergence of vanilla module-wise (both parallel and sequential) training by chaining existing convergence results for shallow modules. We have not considered the impact of the regularization on this, but it is a simple quadratic regularization that makes the target more convex and [29] show the beneficial effects of this transport regularization on training a network end-to-end.
>
> On the theoretical side, assuming therefore that the training of each shallow module converges with the number of epochs, the assumptions under which we have convergence as the regularization weight parameter $\tau$ goes to zero and the number of modules $k$ increases are discussed in Section 2.2 (lines 130 to 136). Convergence as the number of modules increases is the central point in module-wise training as we precisely want the modules to build upon each other to solve the task, and this is exactly what we demonstrate.
>
> In practice, experiments reported on Fig 2 and Fig. 3 show the importance of the regularization term for the convergence to a solution as the number of modules increases.
>
> [A] *Understanding Deep Neural Networks with Rectified Linear Units*, Arora et al., ICLR 2018
>
> [B] *Breaking the Curse of Dimensionality with Convex Neural Networks*, Bach, JMLR 2017
>
> [C] *Beating the Perils of Non-Convexity: Guaranteed Training of Neural Networks using Tensor Methods*, Janzamin et al, arXiv 2016
>
> [D] *Learning One-hidden-layer Neural Networks with Landscape Design*, Ge et al., ICLR 2018
>
> [E] *Improved Learning of One-hidden-layer Convolutional Neural Networks with Overlaps*, Du et al., arXiv 2018
>
> [F] *A Provably Correct Algorithm for Deep Learning that Actually Works*, Malach et al., arXiv 2018
>
> [G] *Provable Bounds for Learning Some Deep Representations*, Arora et al., ICML 2014
>
>
>
> **2. Setting the regularization strength**
>
> It is true that in [28] the early modules are *less biased* towards small displacement values, but regularizing them the same as for deeper layers was beneficial in [29].
>
> We discuss in Section 3.3 and Appendix D the possible need to have a different regularization strength for each module (i.e a different weight $\tau_k$ is given to module $k$), and use an algorithm inspired by the method of multipliers that was also used in [29]  to adjust these values $\tau_k$.
>
> Indeed as you indicate "the selection of the regularization strength becomes more complex when increasing the number of modules $K$", but this is true for all module-wise training methods that add a term to the local loss. The method of multipliers algorithm we use allows us to start all $\tau_k$ from the same initial value and then let them change at different rates during training (Section 3.3 and Appendix D). **This way the number of hyper-parameters required by our method is 3 and does not increase with the number of modules $K$**.
>
> This is more principled than other methods that also add a term to the local loss. For instance, InfoPro [53] have two hyper-parameters per module and they simply assume that they change linearly from the first to the last module and then choose the 4 needed values manually from a fixed set.
>
> Additionally, while observing the behaviour of this algorithm, we noticed that a simple pattern for the values $\tau_k$ works well in most cases and deduced a simple heuristic: use a value $\tau$ for the first $K/2$ modules and double it (i.e use $2 \tau$) for the last $K/2$ modules. A **single hyper-parameter $\tau$** has then to be fixed and this can be done manually or through any search or cross validation method.
>
> **3. Large scale experiments**
>
> Since submitting the paper we have run experiments on ImageNet (ResNet-101 split into 2 modules trained in parallel. Results are below and in the additional PDF file. This validates that the conclusion drawn on the smaller datasets still holds for large datasets such as ImageNet. We did not experiment with more than 2 modules on ImageNet due to resource limitations.
>
> | Dataset  | Parallel VanGL | Parallel TRGL (ours) | DGL   | Sedona | InfoPro | E2E   |
> |----------|----------------|----------------------|-------|--------|---------|-------|
> | ImageNet | 78.11          | **79.41**            | 78.47 | 79.28  | 78.15   | 78.71 |
>
>
> **Questions**
>
> **1. Memory savings**
>
> There might be a misunderstanding here. The memory savings are not less than the other methods. They are most often higher. InfoPro saves more memory in only 2 out of 7 cases (see Tables 4 and 5). InfoProL saves more memory in 2 out of three cases but many of the other methods (Sedona, DDG, FR) don't claim to save memory at all compared to end-to-end training.
>  Please note that VanGL is simply vanilla module-wise training without any added regularization or auxiliary network, so it is normal that it uses less memory than all other methods.
>
> **2. Notations**
>
> Sorry about that, we will add the definition of the pushforward measure $\#$. $T_k$ is simply the module trained in vanilla module-wise training (VanGL), so there is no dependence on $\tau$ which is the weight given to the regularization in TRGL.

---

> ### Comment · Area_Chair_JDVT · 2023-08-21
> **Can you please check the rebuttal comments?**
>
> Dear reviewer,
>
> The authors have provided a response to your comments. Can you please take a look and accordingly comment, and updated your review?
>
> Thanks,
> -Area Chair

---

> ### Comment · Reviewer_U1e3 · 2023-08-21
>
> I want to thank the authors for their detailed response. They addressed my concerns, and I raised my score to weak accept.

---

### Official Review · Reviewer_Ad5i · 2023-07-06

**Soundness:** 2 fair
**Presentation:** 2 fair
**Contribution:** 2 fair
**Rating:** 5
**Confidence:** 3

**Summary:**

TRGL offers a promising approach to module-wise training that addresses the stagnation problem, improves accuracy, and reduces memory usage in constrained and on-device settings. The method introduces a module-wise regularization inspired by the minimizing movement scheme for gradient flows in distribution space. By minimizing the kinetic energy of modules along with the training loss, TRGL encourages modules to change their inputs as little as possible, preserving task-relevant information.

**Strengths:**

(1) TRGL significantly reduces memory usage compared to end-to-end training, ranging from 10% to 60% less memory. This makes it particularly suitable for constrained settings and on-device training where memory resources are limited.

(2) The authors provide theoretical analysis, proving that TRGL leads to more regular and stable greedy modules that progressively minimize the loss.

(3) TRGL can be applied to various network architectures, especially those using residual connections such as ResNets and vision transformers.

**Weaknesses:**

(1) TRGL introduces an additional regularization term to the training objective, which increases the computational complexity of the training process. The calculation of the kinetic energy and its incorporation into the loss function may require additional computational resources, leading to longer training times. How about the actual training time compared to other methods?

(2) what is VanGL? is it just a module-wise training without the regularization terms?

(3) In my opinion, the performance gain is marginal compared  to vanilla VanGL.

(4) The principle of finding Tau is vague.

**Questions:**

See weakness

**Limitations:**

This increased complexity may result in longer training times and higher computational requirements, which could be a limitation in resource-constrained environments or when training large-scale models.

---

> ### Author Rebuttal · Authors · 2023-08-04
>
> Dear Reviewer Ad5i,
>
> Thank you for your valuable review. We answer your remarks below.
>
> **Weaknesses**
>
> **(1) Training time**
>
> Of course like any other method that adds a term to the local loss (so most other methods), training time is slightly increased in favor of saving memory, which is the real hard constraint when training on a small device. We do discuss this in Section 4.2 but probably not prominently enough and quantitatively compare to other methods. Starting line 259:
>
> “Vanilla parallel module-wise (denoted VanGL in the paper) training does slightly slow down training. Epoch time increases by 6% with 2 modules and by 16% with 16 modules compared to end-to-end training. By adding our regularization, TRGL is only slower than VanGL by 2% for any number of modules. This is comparable to InfoPro which reports a time overhead between 1 and 27% compared to end-to-end training. “
>
> Note that InfoPro is the most comparable method to ours in this regard as it also saves memory at the price of slowing down training a little bit, while having performances that are close to or better than end-to-end training. The real limitation for training on mobile devices is memory since the increase in training time is small as stated above: 2% compared to VanGL which has already been used for on-device training in [50,51] as indicated in the paper.
>
>
> **(2) Meaning of VanGL**
>
> Yes this is as you say. “VanGL” is defined on line 209: "We call vanilla greedy module-wise training with the same architecture but without our regularization VanGL, and we include its results in all tables for ablation study purposes"
>
> **(3) Ablation**
>
> The gains are not marginal on parallel training in Section 4.1. They are higher than 1 percentage point in 9 out of the 13 experiments, and higher than 2 percentage points in 6 out of the 13 experiments. This is confirmed by experiments on the larger ImageNet dataset that we added in the general answer to all reviewers above, where the improvement in Top 1 accuracy is 1.3 percentage points.
>
> They are indeed marginal on sequential training in the full data regime, but quite big in the small data regime (up to 6 percentage points gained, Tables 10 and 12 in Appendix G).
>
> Compared to other methods, our experiments cover more cases (parallel and sequential with many or few modules) and show that our method is more reliable as at least it never hurts performance. This is already discussed in Section 5 where we say starting line 294: "The results show that our approach works in all settings (parallel and sequential with many or few modules), whereas other papers don’t test their methods in all settings, and some fail in different settings from the original one in subsequent papers (e.g. delayed gradients methods when the number of modules increases [25] and PredSim in [40]). Also, for parallel training in Section 4.1, the improvement from the regularization compared to VanGL increases with the number of modules (so as the memory savings increase and module-wise training becomes more useful)."
>
> **(4) Choosing hyper-parameter $\tau$**
>
> We experimented with two strategies for choosing the parameter** $\tau$. A method of multipliers is used to adaptively change $\tau$ during training differently for each module and is detailed in Section 3.3 and Appendix D. A reference is made to [29] which also uses this method to adaptively change a regularization weight during training. This is more principled than other methods that also add a term to the local loss. For instance, InfoPro [53] have two hyper-parameters per module and they simply assume that they change linearly from the first to the last module and then choose the 4 needed values manually from a fixed set.
>
> Beside, we also experimented with a simple heuristic that works well in practice and involves setting manually a single hyper-parameter: we use a value $\tau$ for the first $K/2$ modules and double it (i.e use $2 \tau$) for the last $K/2$ modules. So only $\tau$ has to be chosen. Again this is mentioned in Section 3.3.
>
> **Limitations**
>
> The real limitation for training on mobile devices is memory since the increase in training time is small as stated above: 2% compared to VanGL which has already been used for on-device training [50,51] as indicated in the paper.

---

> ### Comment · Area_Chair_JDVT · 2023-08-21
> **Can you please check the rebuttal comments?**
>
> Dear reviewer,
>
> The authors have provided a response to your comments. Can you please take a look and accordingly comment, and updated your review?
>
> Thanks,
> -Area Chair

---

### Author Rebuttal · Authors · 2023-08-05

Thanks to all the reviewers for their valuable insights. We have addressed in the individual answers the points raised by the reviewers. Since some comments are shared by different reviewers, we summarise here the main answers:

**1. Experiments on ImageNet (Reviewers U1e3 and N1H7)**

The reviewers asked for experiments on larger datasets. Since submitting the paper we have run experiments on ImageNet (ResNet-101 split into 2 modules trained in parallel). Results are below and in the additional PDF file. This validates that the conclusion drawn on the smaller datasets still holds for large datasets such as ImageNet. We did not experiment with more than 2 modules on ImageNet due to resource limitations.

| Dataset  | Parallel VanGL | Parallel TRGL (ours) | DGL   | Sedona | InfoPro | E2E   |
|----------|----------------|----------------------|-------|--------|---------|-------|
| ImageNet | 78.11          | **79.41**            | 78.47 | 79.28  | 78.15   | 78.71 |

**2. Training time (Reviewers Ad5i and N1H7)**

The reviewers asked about the increase in wall-time training cost caused by our regularization. This was included in the paper but not prominently enough. In section 4.2 we say:

Vanilla parallel module-wise (denoted VanGL in the paper) training does slightly increase training time. Epoch time increases by 6% with 2 modules and by 16% with 16 modules compared to end-to-end training. By adding our regularization, TRGL is only slower than VanGL by 2% for any number of modules. This is comparable to InfoPro baseline which reports a time overhead between 1 and 27% compared to end-to-end training.

Note that InfoPro is the most comparable method to ours in this regard as it also saves memory at the price of slowing down training a little bit, while having performances that are close to or better than end-to-end training.


**3. Convergence (Reviewers U1e3 and X5WU)**

Indeed this is an important question and this will be further discussed in the final version. In Section 2.2, we have assumed that the training of the individual modules converges.

Indeed, the training of shallow networks is known to converge [A,B,C,D,E]. Papers [5,6,24,F,G] show the convergence of vanilla module-wise (both parallel and sequential) training by chaining existing convergence results for shallow modules. We have not considered the impact of the regularization on this, but it is a simple quadratic regularization that makes the target more convex and [29] show the beneficial effects of this transport regularization on training a network end-to-end.

Assuming therefore that the training of each shallow module converges with the number of epochs, the assumptions under which we have convergence as the regularization weight parameter $\tau$ goes to zero and the number of modules (denoted $k$ ln 131) increases are discussed in Section 2.2 (lines 130 to 136). Convergence as the number of modules increases is the central point in module-wise training as we precisely want the modules to build upon each other to solve the task, and this is exactly what we demonstrate.

In practice, experiments reported on Fig 2 and Fig. 3 show the importance of the regularization term for the convergence to a solution as the number of modules increases.

[A] *Understanding Deep Neural Networks with Rectified Linear Units*, Arora et al., ICLR 2018

[B] *Breaking the Curse of Dimensionality with Convex Neural Networks*, Bach, JMLR 2017

[C] *Beating the Perils of Non-Convexity: Guaranteed Training of Neural Networks using Tensor Methods*, Janzamin et al, arXiv 2016

[D] *Learning One-hidden-layer Neural Networks with Landscape Design*, Ge et al., ICLR 2018

[E] *Improved Learning of One-hidden-layer Convolutional Neural Networks with Overlaps*, Du et al., arXiv 2018

[F] *A Provably Correct Algorithm for Deep Learning that Actually Works*, Malach et al., arXiv 2018

[G] *Provable Bounds for Learning Some Deep Representations*, Arora et al., ICML 2014


**4. Choosing hyper-parameter $\tau$ (Reviewers Ad5i and U1e3)**

We discuss in Section 3.3 and Appendix D the possible need to have a different regularization strength for each module (i.e a different weight $\tau_k$ is given to module $k$), and use an algorithm inspired by the method of multipliers (that was also used in [29]) to adjust these values $\tau_k$.

The method of multipliers algorithm we use allows us to start all values $\tau_k$ from the same initial value and then let them change at different rates during training. **This way the number of hyper-parameters required by our method is 3 and does not increase with the number of modules $K$**.

This is more principled than other methods that also add a term to the local loss. For instance, InfoPro [53] have two hyper-parameters per module and they simply assume that they change linearly from the first to the last module and then choose the 4 needed values manually from a fixed set.

Additionally, while observing the behaviour of this algorithm, we noticed that a simple pattern for the values $\tau_k$ works well in most cases and deduced a simple heuristic: use a value $\tau$ for the first $K/2$ modules and double it (i.e use $2 \tau$) for the last $K/2$ modules. A **single hyper-parameter $\tau$** has then to be fixed and this can be done manually or through any search or cross validation method.

**5. Evidence for early overfitting (Reviewer N1H7)**

Besides our experiments in Figures 2 and 3, we cite 4 papers [36,5,53,40] on line 33 that make the same observation. The arguments developed in each paper are detailed below in the individual answer to Reviewer N1H7. We therefore considered that this was well-established in the literature. We propose to add the discussion in the individual answer to Reviewer N1H7 of the evidence in the literature for this phenomenon to the paper.

**6. Further sensitivity to hyper-parameter experiment (Reviewer X5WU)**

See Appendix H and a new experiment in the additional PDF

---

### Author Response · Authors · 2023-08-18
**Dear reviewers**

Dear reviewers, thank you once again for your valuable comments and insightful recommendations.

As the deadline for the discussion approaches, we would like to gently remind you of our eagerness to receive your response and engage in further discourse regarding the points raised in the rebuttal.

We would like to draw your attention to the fact that one of the primary weaknesses identified was the lack of comprehensive large-scale evaluation. In addressing this concern, we have conducted additional experiments and are pleased to present new quantitative results obtained from large datasets (imageNet). These results serve to reinforce and substantiate the conclusions we initially put forth.

Thanks again,
Authors

---

### Decision · Program_Chairs · 2023-09-21

**Decision:**

Accept (poster)

**Comment:**

The submission proposes a training method for greedy layer-/module- wise training of neural networks. In contrast to standard (full-network) backpropagation, which has the requirement of keeping the full network and computation graph in the memory during the training, module-wise training relies on chopping up the network into modules that can be trained more independently. This is useful in resource-bounded settings such as on-device training, but the approximation comes with certain endemic issues. The goal of the submissions is to alleviate some of them.

All of the reviewers recommend the submission for acceptance. The rebuttal was particularly useful and changed the opinion of several of the reviewers toward acceptance. The newly reported results on larger-scale experiments were also found helpful. The authors are recommended to incorporate the comments raised during the review process in preparing their camera-ready.